# Augmented Bone Morphogenetic Protein Signaling During TMJ Development Alters Morphology in a Timepoint-Dependent Manner

**DOI:** 10.3390/ijms26041655

**Published:** 2025-02-15

**Authors:** Susannah C. Midla, Maiko Omi-Sugihara, Madeline Cha, Coral Chen, Rafael Correia Cavalcante, Haichun Pan, Yuji Mishina, Hiroki Ueharu

**Affiliations:** 1Department of Biologic and Materials Sciences & Prosthodontics, University of Michigan School of Dentistry, Ann Arbor, MI 48109-1078, USA; smidla@umich.edu (S.C.M.); sugihara.maiko.dent@osaka-u.ac.jp (M.O.-S.); madecha@umich.edu (M.C.); coralchen5056@gmail.com (C.C.); rafaelcc@umich.edu (R.C.C.); haichun@umich.edu (H.P.); 2Department of Orthodontics and Dentofacial Orthopedics, Graduate School of Dentistry, Osaka University, Osaka 565-0871, Japan

**Keywords:** condylar cartilage, BMPR1A, cranial neural crest

## Abstract

The temporomandibular joint (TMJ) is unique in both developmental origin and functional maintenance. The role of bone morphogenic protein (BMP) signaling in endochondral ossification has been widely investigated but not in the context of the TMJ. We employed a histomorphometric analysis approach to understand how augmented BMP signaling in the cranial neural crest affects the postnatal development of the TMJ. Our analysis showed that cartilage length in the mandibular condyle was reduced in *Wnt1 Cre;caBmpr1a* mice before the weaning stage (P17). However, following weaning, the mandibular condylar cartilage showed recovered length (P28 and P42). Furthermore, the changes in cartilage length coincide with alterations in cell death in the superficial region of the mandibular condyle. These results suggest that BMP signaling influences chondrocyte cell death and TMJ development in a timepoint-specific manner.

## 1. Introduction

The temporomandibular joint (TMJ) is one of the most unique and important joints in the body and is critical to overall health because of its role in mastication and swallowing [1,2]. The TMJ is a synovial joint composed of three main components: the mandibular condyle, the disc, and the temporal bone glenoid fossa [3,4]. The TMJ is functionally important for speaking, mastication, and breathing [5,6]. During development, the secondary cartilage of the mandibular condyle extends the mandible posteriorly through endochondral ossification [7,8]. The development of the mandibular condyle contributes to a major portion of the posterior mandible [9]. Alternatively, the anterior portion of the mandible develops via intramembranous ossification from the ectomesenchyme neural crest cell population [10]. Endochondral ossification and intramembranous ossification are two distinct processes that result in the formation of bone. Both processes begin with condensation of mesenchymal cells; however, during endochondral ossification, cells differentiate first into chondrocytes then osteoblasts, while intramembranous ossification cells differentiate directly into osteoblasts [11,12]. The TMJ is therefore unique in its developmental origin. Bone morphogenetic proteins (BMPs) play a critical role in bone development and were originally discovered due to their ability to induce ectopic bone formation [13]. BMP signaling occurs through BMP ligand binding to heterodimers of type I and type II receptors and subsequent Smad protein phosphorylation [14]. BMP signaling is crucial for endochondral ossification, as deletion of *Bmpr1a* from chondrocytes using Aggrecan-Cre results in the failure of chondrocyte-derived osteoblast and osteocyte formation [15]. Lineage progression through different chondrocyte populations and mineral deposition during endochondral ossification is essential for TMJ development and normal mandibular condyle formation [16,17]. BMPs function as a promoter of chondrogenesis during endochondral ossification from the mesenchymal condensation stage to the differentiation of hypertrophic chondrocytes in the growth plate [18]. This has been demonstrated both in vitro [19,20] and in vivo [21]. BMP signaling is critical both during embryonic development and adult homeostasis of stem cell populations. Our previous work has shown that the elevation of BMP signaling through a constitutively active mutant form of the BMPR1a, a type I receptor for BMPs, in neural crest cells can induce ectopic cartilage formation in the cranial sutures of mice [22,23]. Cranial neural crest cells (CNCCs) are multipotent migrating mesenchymal cells that give rise to ectodermal cell populations, including osteoblasts, chondrocytes, and adipocytes [24,25,26,27]. Our previous works have shown that augmentation of BMP signaling in CNCCs alters their cell fate towards chondrogenic lineage [22]. The ectopic cartilage in the cranial sutures leads to suture fusion or craniosynostosis in these mice. Alterations in both overall skull morphology due to craniosynostosis in these mice and the inherent role of endochondral ossification in TMJ development led us to investigate the TMJ of mice with augmented BMP signaling in CNCCs. In this study, we analyzed the impact of elevated BMP signaling in neural crest cells on endogenous cartilage formation in the TMJ.

## 2. Results

### 2.1. Wnt1 Cre;caBmpr1a Mice Show a More Severe Mandibular Phenotype Compared to P0 Cre;caBmpr1a Mice

To investigate the impact of augmented BMP signaling in neural crest cells on TMJ development, we crossed transgenic mice that express a constitutively active form of Bmpr1a (caBmpr1a mice) with two widely used neural crest-specific Cre-lines, *Wnt1 Cre* [28] and *P0 Cre* [29], to generate *Wnt1 Cre;caBmpr1a* and *P0 Cre;caBmpr1a* mutant mice. Both *Wnt1 Cre;caBmpr1a* mice and *P0 Cre;caBmpr1a* mice exhibited higher amounts of pSMAD1/5/9 in cranial NCCs compared to controls [22]. These mice have also been previously characterized with short, rounded faces and craniosynostosis [22,23]. We carried out in-depth characterization of the shape of the mandible at postnatal day 17 (P17) utilizing μCT images and previously established landmarks [30,31,32,33] (Figure 1A,B). There was a significant reduction in the anterior length, ascending height, descending height, and posterior height of the *Wnt1 Cre;caBmpr1a* mice when compared to the controls (Figure 1B,C). *P0 Cre;caBmpr1a* mice showed a non-significant trend towards smaller mandibles (Figure 1B,C); however, because of the less severe phenotype, we proceeded with *Wnt1 Cre;caBmpr1a* mice for the rest of the analysis. The condyle length of the TMJ was similar across all three genotypes (Figure 1C).

### 2.2. Wnt1 Cre;caBmpr1a Mice Have Smaller Mandibular Condyles Through Development That Are Recovered Post-Weaning

To assess the development of the TMJ in wild-type and *Wnt1 Cre;caBmpr1a* mice, we harvested at P3, P7, P17, P28, and P42 and analyzed the morphological changes via H&E staining. The overall shape of the TMJ was consistent at each developmental stage between control and *Wnt1 Cre;caBmpr1a* mice (Figure 2A). As established previously, we used the cell shape, size, and level of staining to delineate four distinct zones in the mandibular condyle: superficial, proliferative, pre-hypertrophic, and hypertrophic zones [30,34]. In this measurement, we combined the superficial and proliferative zones together. Briefly, we used the most anterior and posterior points of the condyle to draw a line, then analyzed the length of each region at the point of tangency of a line parallel to the anterior–posterior line. We analyzed the total length and three zones of the mandibular condyle to assess changes in TMJ morphology of control and *Wnt1 Cre;caBmpr1a* mice (Figure 2B). There was a significant reduction in the total length of *Wnt1 Cre;caBmpr1a* mandibular condyles at developmental stages prior to weaning, P3, P7, and P17 (Figure 2C). Furthermore, at P3, there was a significant reduction in the pre-hypertrophic zone of the *Wnt1 Cre;caBmpr1a* mandibular condyles (Figure 2C). At P7, the hypertrophic zone of *Wnt1 Cre;caBmpr1a* mandibular condyles showed a significant reduction in length (height) (Figure 2C). At P17, the *Wnt1 Cre;caBmpr1a* mandibular condyles had a significant reduction in all distinct chondrocyte zones (Figure 2C). In contrast, the stages analyzed following weaning showed different phenotypes. At P28, there was no significant difference in total length or any chondrocyte zone between control and *Wnt1 Cre;caBmpr1a* mandibular condyles (Figure 2C). At P42, however, *Wnt1 Cre;caBmpr1a* mice showed an longer overall length of the mandibular condyle cartilage and a larger hypertrophic zone length (Figure 2C). Over time, the total length of the mandibular condyle cartilage decreased in both control and *Wnt1 Cre;caBmpr1a* mice (Figure 2D).

### 2.3. BMP Signaling in Neural Crest Cells Does Not Impact Extracellular Matrix Composition in the TMJ

To assess the quality of the extracellular matrix of control and *Wnt1 Cre;caBmpr1a* mandibular condyles, we carried out Safranin O staining for glycosaminoglycans (GAGs) [35]. Throughout development, the mandibular cartilage of both control and *Wnt1 Cre;caBmpr1a* mice showed similar levels of red color indicating that there was no change in the GAG content between genotypes (Figure 3).

### 2.4. Cell Death and Proliferation Lead to Alterations in Wnt1 Cre;caBmpr1a Condylar Cartilage Throughout Development

To understand the processes mediating changes in *Wnt1 Cre;caBmpr1a* mandibular condyle lengths, we investigated the level of cell death through the TUNEL assay and cell proliferation with phospho-histone H3 immunohistochemistry (Figure 4A). We analyzed the superficial zone and pre-hypertrophic zone together and the hypertrophic zone separately (Figure 4B). The superficial zone (Sp) is defined as areas that have low or no Safranin O stain, which includes superficial, proliferative, and prehypertrophic zones, and the hypertrophic zone (Hyp) as the area that has a dark Safranin O stain (Figure 4B). There was no difference in proliferation or cell death at P3 or P7 in either the superficial or hypertrophic zone of the *Wnt1 Cre;caBmpr1a* mandibular condyles (Figure 4C). There was a significant upregulation in cell death in the superficial zone of P17 mandibular condyles (Figure 4C). Additionally, we observed no significant difference in P28 or P42 cell death or proliferation in either the superficial or hypertrophic zone. There was a non-significant trend towards reduced cell death in the superficial zone of P42 *Wnt1 Cre;caBmpr1a* mandibular condyles.

## 3. Discussion

In this study, we utilized *Wnt1 Cre;caBmpr1a* mice to assess the effect of elevated BMP signaling in the neural crest on TMJ development. From our histomorphometric analysis, we found that BMP signaling in the neural crest influences the development of the mandibular condyle by altering cell death in superficial (proliferative) chondrocytes. Elevated BMP signaling in the neural crest resulted in a shorter mandibular condyle cartilage length at developmental stages prior to weaning. However, following weaning, the effect of elevated BMP signaling was progressively reversed with mandibular condyle cartilage length to reverse the phenotype in *Wnt1 Cre;caBmpr1a* mice at P42.

Previous work has shown that conditional *Bmpr1a* knockout in chondrocytes reduces the size of the TMJ during postnatal development from postnatal day 3 to four weeks old [36]. Additional studies have also shown TMJ hypoplasia in mouse embryos following the disruption of Bmpr1a in the neural crest [37]. Interestingly, this study also found degeneration of the TMJ entirely at E18.5 when BMPR1A signaling was augmented in a neural crest-specific manner using their version of *caBmpr1a* mice [37]. The authors did not investigate later timepoints than E18.5 due to the perinatal lethality. It is possible that the mutant mouse line demonstrates much higher levels of BMPR1A signaling activities than the mice used in these studies. The mice used in this study were heterozygous for *caBmpr1a*, and it is possible to speculate that homozygous mice for *caBmpr1a* may show a different phenotype. However, previous work by our group has shown that there are no significant differences in long bone phenotypes between mice heterozygous or homozygous for the *caBmpr1a* gene [38].

The current results suggest that the alteration of BMP signaling can affect TMJ morphology in a timepoint-dependent manner; thus, early alterations in morphology may eventually be reversed. Previous work depleting BMP signaling in mature osteoblasts has shown reduced bone mass in young mice but more bone mass in aged mice [39]. This indicates that BMP signaling has distinct roles at different development stages. Our study does not investigate the role of BMP signaling in the maintenance of the TMJ. This would be an important future direction as TMJ disorders, including osteoarthritis, affect as many as 36 million adults in the US [40]. Age should be taken into consideration when studying the role of BMP signaling in cartilage as we have found differential effects based on age.

In this study, we found excess cell death during the TMJ development in a spatial and temporal manner (Figure 4). Our previous study using the same transgenic mouse line demonstrated that enhanced BMP signaling induces p53-mediated cell death in the nasal cartilage and synchondrosis [41,42]. Our other transgenic mouse line to enhance BMP signaling in cranial NCCs (*P0 Cre;caAcvr1* mice) also elevates the p53-mediated cell death in the nasal process at the early embryonic stage [43]. These data suggest that the excess cell death found in the mutant TMJ is also mediated by p53. However, it is still unclear how the BMP/p53-induced cell death is spatially and temporally regulated during the TMJ development.

Interestingly, the reversal of mandibular condyle cartilage length (Figure 2D) coincides with the transition from the soft diet of milk to hard chow following weaning (P17). Immobilization of the mandible has been shown to lead to cartilage thinning and degeneration both in rabbits [44] and primates [45]. However, upon remobilization, there is some recovery of GAGs and cartilage structure [44,45]. Furthermore, rats fed a soft diet show also show thinner articular cartilage, shorter mandibular ramus height, and lowered chondroblast differentiation marker expression [46,47]. Our previous work has shown that a soft diet in the osteoblast-specific disruption of BMP signaling reduces the impact of BMP signaling loss on mandibular condyle cartilage morphology [30]. This study has also shown that the anterior and posterior regions of the mandibular condyle respond differently to mechanical force [30]. We only investigated the central portion of the mandibular condyle, and therefore the response of the anterior and posterior portions of the mandibular condyle during development remains an area of investigation.

Notably, mice with osteoblast-specific loss of *Bmpr1a* respond to treadmill exercise to increase bone mass [48]. These results suggest an interplay between BMP signaling and mechanosensing pathways. Mechanosensing has recently been highlighted during bone development and craniofacial development. Transcription activators for mechanosensing, YAP and TAZ, govern osteoprogenitor maintenance and subsequent differentiation with BMP signaling [49,50,51]. Neural crest-specific deletions of *Yap* and *Taz* (*Wnt1 Cre;Yap^+/−^;Taz^−/−^* cKO mice) develop ectopic cartilage in the frontal suture leading to premature suture closure [52]. Remarkably, our *Wnt1* and *P0 Cre;caBmpr1a* mice also develop ectopic cartilage in the frontal suture that leads to premature suture closure [22]. These data suggest that mechanosensing is a critical factor contributing to BMP-induced differentiation toward either osteogenic or chondrogenic lineages depending on the presence or absence of YAP/TAZ. Another aspect is mechanosensing as a conductor for BMP-induced cell death. Neural crest-specific deletion of *Bmpr1a* reduces cell death in the interzone between the condyle and the glenoid fossa [37], suggesting an essential role of BMP-induced cell death for cavitation in the TMJ. Interestingly, hedgehog signaling, a transducer of mechanosensing, upregulates BMP signaling in the TMJ [37,53]. These data suggest that changes in mechanical forces through the transition from the soft diet of milk to hard chow affect either or both cell fate specification and cell death by BMP signaling. However, the role of mechanical force and development in the TMJ are hard to separate in the current study. The relationship between BMP signaling, mechanical force, and chondrogenesis in the TMJ remains unclear and warrants future efforts. This study shows that BMP signaling influences the proliferation and cell death of chondrocytes in the TMJ in a developmental-stage-dependent manner.

## 4. Materials and Methods

### 4.1. Mice

To augment BMP signaling in cranial neural crest cells, *Wnt1 Cre* mice or *P0 Cre* mice were bred with mice carrying the constitutively active form of Bmpr1a (*caBmpr1a*) that has been previously described [23]. *P0 Cre* mice (C57BL/6JTg(P0-Cre)94Imeg (ID 148)) were provided by CARD, Kumamoto University, Kumamoto, Japan [29]. *Wnt1-Cre* mice (B6.Cg-Tg(Wnt1-cre)11Rth Tg(Wnt1-GAL4)11Rth/J) were obtained from Jaxson laboratory Stock# 009107) [28]. Mice genotyped carrying *Wnt1 Cre* and *caBmpr1a* are referred to as *Wnt1 Cre;caBmpr1a*, and mice genotyped carrying *P0 Cre* and *caBmpr1a* are *P0 Cre;caBmpr1a*. All mice were fed a regular rodent diet and housed in pathogen-free group housing. All animal experiments were carried out in accordance with the Institutional Animal Care and Use Committee (IACUC) laws and policies at the University of Michigan (PRO00009613, approved on 24 April 2023).

### 4.2. Micro-Computed Tomography

Micro-CT scanning of fixed heads was performed at the University of Michigan in collaboration with the micro-CT core (μCT100 Scanco Medical, Bassersdorf, Switzerland). Scan settings were performed with the following settings: a voxel size of 12 μm, 55 kVp, 109 μA, 0.5 mm AL filter, and an integration time of 500 ms. ITK-SNAP was utilized to generate surface models from CT scans [54,55]. ITK-SNAP was used for image analysis, and 3D Slicer was used for image production. Anatomical landmarks were placed according to those shown in Figure 1B.

### 4.3. Histology and Immunohistochemistry

Mouse heads were harvested at postnatal day 3 (P3), postnatal day 7 (P7), postnatal day 17 (P17), postnatal day 21 (P21), and postnatal day 42 (P42) and fixed overnight at 4 °C in 4% paraformaldehyde (PFA). These timepoints were chosen to assess the progression of TMJ development postnatally with a focus on the weaning stages. P17, P21, and P42 heads were immersed in 14% EDTA in PBS pH 7.4 for 14 days then in 30% sucrose in PBS until the tissues sank. P3 and P7 heads were immersed in 30% sucrose in PBS until the tissues sank. Heads were subsequently embedded in optimal cutting temperature compound (OCT, Thermo Fisher Scientific, Waltham, MA, USA) and sectioned at 10 μM thickness using the LeicaCM1850 cryostat (Leica, Wetzlar, Germany). To observe the condyle morphology, Hematoxylin and Eosin (H&E) and Safranin O assays were performed according to standard protocols. Histomorphometric analyses of the condylar cartilage were conducted using ImageJ (ver. 1.53a). The mandibular condylar cartilage was divided into superficial (fibrous), proliferative, prehypertrophic, and hypertrophic cell layers based on the cell shape, size, and staining intensity [30,34]. The terminal deoxynucleotidyl transferase dUTP nick end-labeling (TUNEL) assay was performed using the Click-ITTM Plus TUNEL assay (#C10617, Invitrogen, Waltham, MA, USA). Phospho-histone H3 (pH3) immunohistochemistry was carried out with 1:200 phosphor-histone H3 (Ser10) polyclonal antibody (#44-1190G, Invitrogen, Waltham, MA, USA) and donkey anti-rabbit 488 secondary antibody. Fluorescence images were taken by confocal microscopy (Eclipse Ti, Nikon, Melville, NY, USA).

### 4.4. Experimental Groups

μCT analysis involved *n* = 3 samples per genotype. H&E staining included *n* = 4 mice per genotype per timepoint and 3 sections corresponding to the center of the mandibular condyle per mouse analyzed. pH3/TUNEL involved *n* = 3 samples per genotype per timepoint and 2 sections corresponding to the center of the mandibular condyle per mouse analyzed.

### 4.5. Statistical Analysis

Statistical analyses were carried out using GraphPad Prism 10.0 software. For the comparison of two groups of mice, Student’s *t*-test was performed. For comparison of more than two groups of mice, one-way ANOVA with Turkey’s test correction for multiple comparisons was performed. Significance indicated by asterisks: * *p* < 0.05, ** *p* < 0.01.

## Figures and Tables

**Figure 1 ijms-26-01655-f001:**
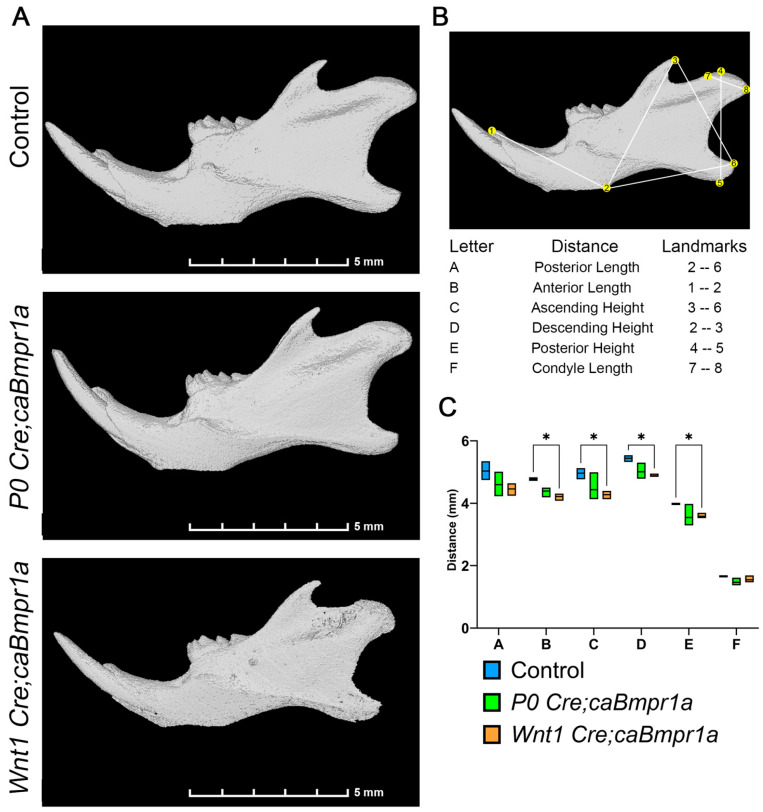
*Wnt1 Cre;caBmpr1a* mice show a more severe mandibular phenotype than *P0 Cre;caBmpr1a* mice. (**A**) Representative images of μCT images of control and mutant mandibles. Scale bar = 5 mm. (**B**) Key of landmarks used for morphological analysis of mandible μCT. (**C**). Analysis of mandible lengths reveals that anterior length, ascending height, descending height, and posterior height are significantly shorter in *Wnt1 Cre;caBmpr1a*. No significant changes were observed for *P0 Cre;caBmpr1a*. No significant changes in posterior length or condyle length were recorded for any mutant. Scale = 5 mm, *n* = 3, * *p* < 0.05.

**Figure 2 ijms-26-01655-f002:**
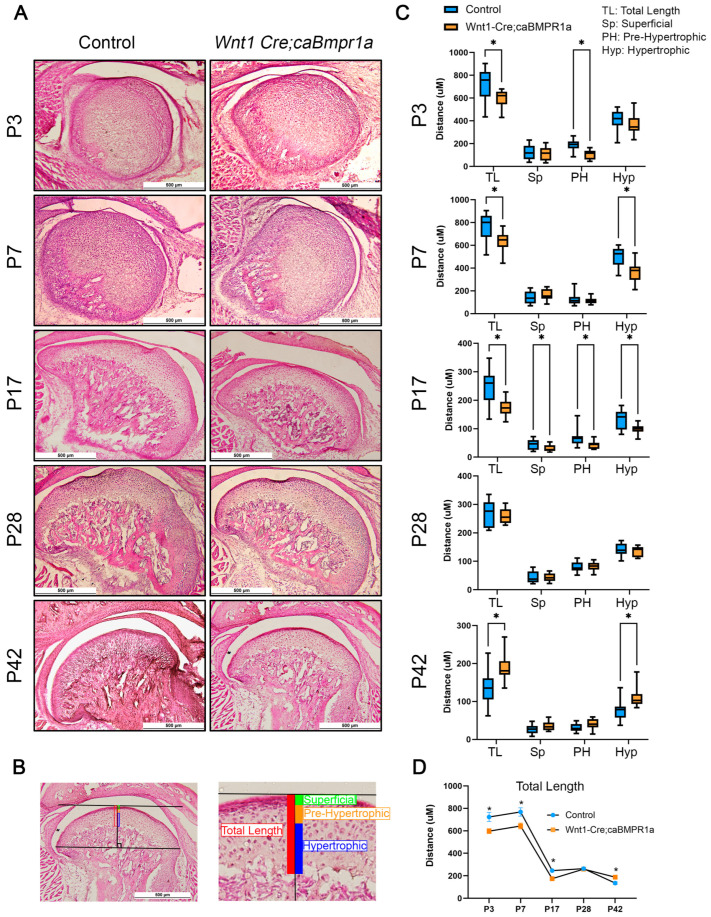
Changes in the histomorphometric analysis of *Wnt1 Cre;caBmpr1a* mice are timepoint-dependent. (**A**) Representative images of H&E staining of P3, P7, P17, P28, and P42 mandibular condyle sagittal sections. (**B**) Key of different cellular zones in the mandibular condylar cartilage. (**C**) Analysis of condylar cartilage length showing a significantly smaller total length at P3, P7, and P17. TL: total length, Sp: superficial region (a combination of superficial and proliferative zones), PH: pre-hypertrophic region, Hyp: hypertrophic region. No significant changes were observed at P28. Significantly higher total length at P42. (**D**) Analysis of the total length of the condylar cartilage across different timepoints showing a decrease in cartilage length in both control and mutant throughout development. Scale = 500 mm, *n* = 4, * *p* < 0.05.

**Figure 3 ijms-26-01655-f003:**
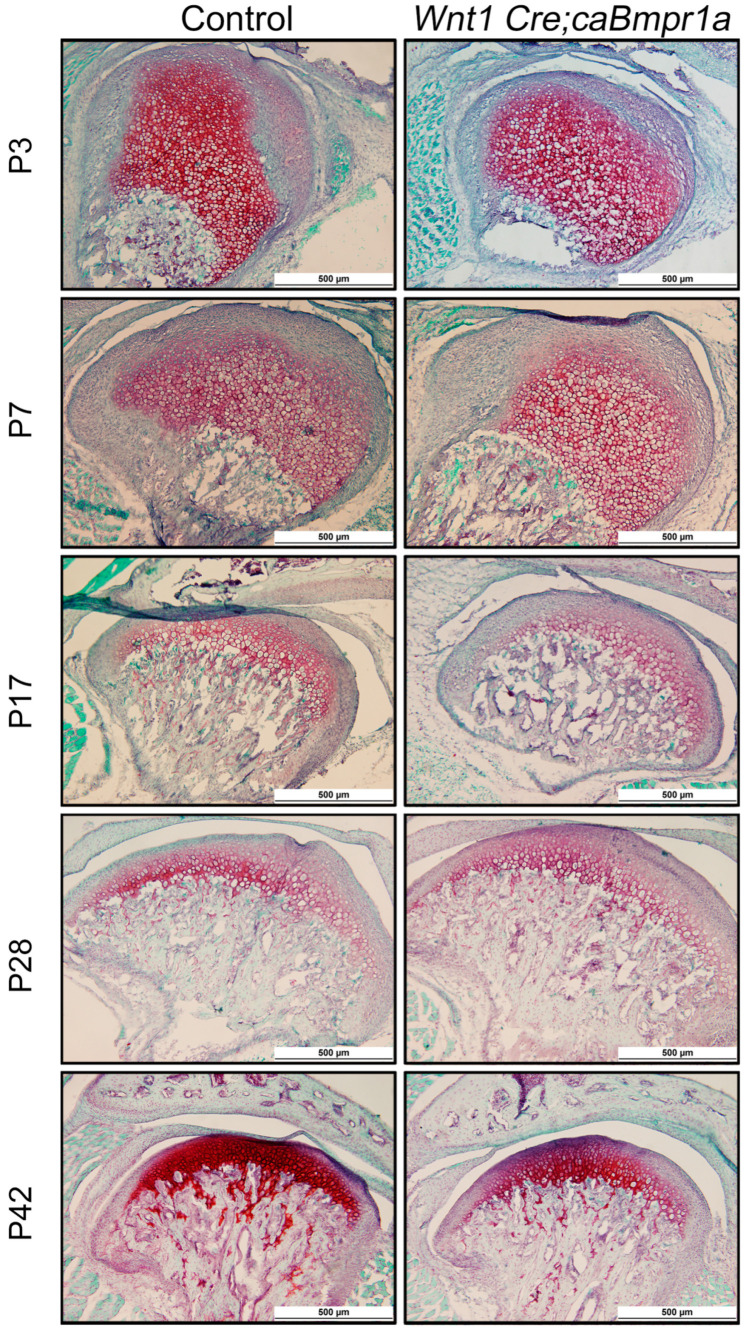
Extracellular matrix remains unchanged between control and *Wnt1 Cre;caBMPR1a* mandibular condylar cartilage. Representative images of control and mutant mice stained with Safranin O and Fast Green at P3, P7, P17, P28, and P42. Scale = 500 mm, *n* = 3.

**Figure 4 ijms-26-01655-f004:**
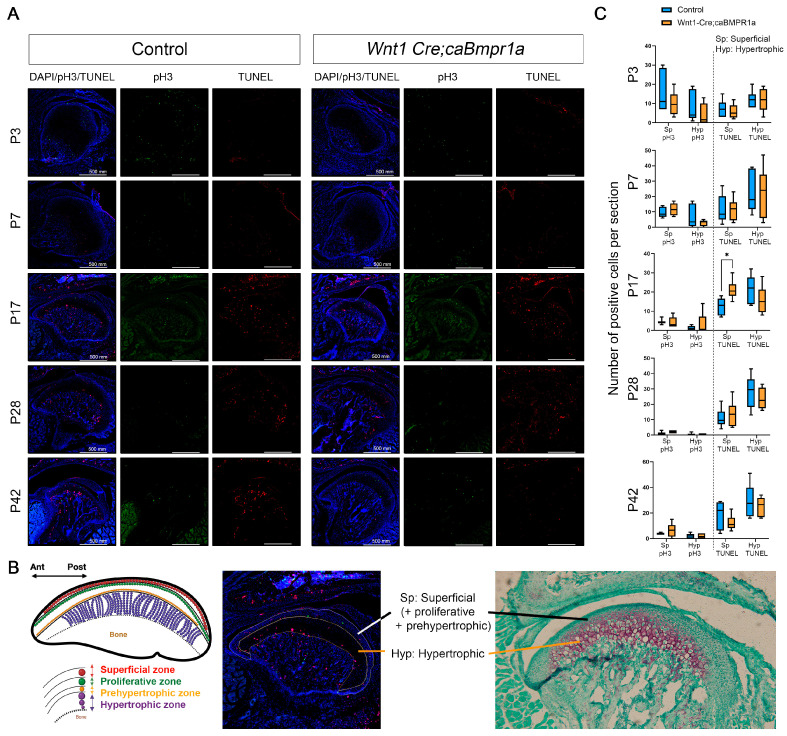
Cell death in the superficial region coincides with alterations in total mandibular cartilage length. (**A**) Representative images of DAPI, phospho-histone H3, and TUNEL staining at P3, P7, P17, P28, and P42 of control and *Wnt1 Cre;caBmpr1a* mandibular condyle. (**B**) Representation of superficial (a combination of superficial, proliferative, and prehypertrophic zones) and hypertrophic cellular zones used for analysis. (**C**) There is a significantly larger level of cell death in the superficial zone of P17 *Wnt1 Cre;caBmpr1a* condylar cartilage. No other cellular zones or developmental stages show significant differences in cell death or cell proliferation. Scale = 500 mm, *n* = 3, * *p* < 0.05.

## Data Availability

Data is contained within the article.

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
