# Peer review of "Augmented Bone Morphogenetic Protein Signaling During TMJ Development Alters Morphology in a Timepoint-Dependent Manner"

_ijms, 2025, doi:10.3390/ijms26041655_

Round 1

Reviewer 1 Report

Comments and Suggestions for Authors

This study the effect of enhanced BMP signaling in cranial neural crest cells (CNCCs) on the developmental anatomy of the temporomandibular joint (TMJ). This represents a new and relatively unexplored frontier in craniofacial biology, considering the important roles played by BMP signaling in both chondrogenesis, endochondral ossification and skeletal remodeling.Tools to study IFNB09 pathway regulation in chondrocytes will complement both existing models to study the endochondral ossifications that underpin the craniofacial skeleton, and current skeletal remodeling models. The authors present an investigation of developmental dynamics and time-dependent effects of BMP signaling in transgenic mouse models (Wnt1 Cre;caBmpr1a and P0 Cre;caBmpr1a), which is likely of great importance to the fields of developmental biology, orthopedics, and TMJ disorders.

The methodology of the study is broad, encompassing, and well put-together. To promote BMP signaling specifically in cranial neural crest cells, the researchers created two transgenic mouse lines (Wnt1 Cre;caBmpr1a and P0 Cre;caBmpr1a). They afforded point-by-point analyses at multiple developmental time points (P3 through P42), which importantly encompass pre- and post-weaning periods. In doing so, this time-ordered analysis was critical to revealing the temporal nature of BMP signaling responses.

First, studies using Wnt1 Cre;caBmpr1a mice showed drastic reductions in mandibular dimensions as compared to controls, with a more severe phenotype than previously seen with the P0 Cre;caBmpr1a mice. Thus, this difference in the two Cre lines demonstrates specificity regarding how distinct neural crest populations contribute to TMJ development.

Secondly, they found a remarkable temporal pattern in the development of the mandibular condyle cartilage. Before the weaning stage (P3-P17), the length of cartilage in the mutant mice was significantly less pronounced. However, post-weaning (P28-P42), it demonstrated significant recovery. This discovery indicates a novel plasticity of TMJ development with compensatory effects.

Third, at the cellular level, they noted a higher rate of cell death in the superficial zone at P17, which coincided with the most severe phenotype. The temporal associations between cell death and morphological changes offer insight into cellular control mechanisms resulting in TMJ formation.

This study concluded that BMP signaling seems to be a critical time-dependent regulator in TMJ development, particularly influencing cell death, specifically highlighted in superficial chondrocytes. This research demonstrates an unanticipated temporal dimension of BMP signaling in TMJ formation, which has differential effects pre- and post- weaning.

A number of weaknesses, however, deserve mention:

The most notable limitation is that, however, they rarely investigated molecular mechanisms. In contrast, though the study clearly shows augmented BMP signaling affects TMJ development, it does not discuss how this occurs at the molecular level. Identifying which of the downstream targets of BMP signaling mediate these effects would greatly strengthen these findings. It remains unknown during post-weaning recovery which compensatory pathways are activated. What triggered this recovery? If the authors include the points in the discussion, the manuscripts will be better. And make their analyses of the clinical implications strongerplease fix some grammar errors.

Author Response

Comments from Reviewer1

This study the effect of enhanced BMP signaling in cranial neural crest cells (CNCCs) on the developmental anatomy of the temporomandibular joint (TMJ). This represents a new and relatively unexplored frontier in craniofacial biology, considering the important roles played by BMP signaling in both chondrogenesis, endochondral ossification and skeletal remodeling. Tools to study IFNB09 pathway regulation in chondrocytes will complement both existing models to study the endochondral ossifications that underpin the craniofacial skeleton, and current skeletal remodeling models. The authors present an investigation of developmental dynamics and time-dependent effects of BMP signaling in transgenic mouse models (Wnt1 Cre;caBmpr1a and P0 Cre;caBmpr1a), which is likely of great importance to the fields of developmental biology, orthopedics, and TMJ disorders.

The methodology of the study is broad, encompassing, and well put-together. To promote BMP signaling specifically in cranial neural crest cells, the researchers created two transgenic mouse lines (Wnt1 Cre;caBmpr1a and P0 Cre;caBmpr1a). They afforded point-by-point analyses at multiple developmental time points (P3 through P42), which importantly encompass pre- and post-weaning periods. In doing so, this time-ordered analysis was critical to revealing the temporal nature of BMP signaling responses.

First, studies using Wnt1 Cre;caBmpr1a mice showed drastic reductions in mandibular dimensions as compared to controls, with a more severe phenotype than previously seen with the P0 Cre;caBmpr1a mice. Thus, this difference in the two Cre lines demonstrates specificity regarding how distinct neural crest populations contribute to TMJ development. Secondly, they found a remarkable temporal pattern in the development of the mandibular condyle cartilage. Before the weaning stage (P3-P17), the length of cartilage in the mutant mice was significantly less pronounced. However, post-weaning (P28-P42), it demonstrated significant recovery. This discovery indicates a novel plasticity of TMJ development with compensatory effects. Third, at the cellular level, they noted a higher rate of cell death in the superficial zone at P17, which coincided with the most severe phenotype. The temporal associations between cell death and morphological changes offer insight into cellular control mechanisms resulting in TMJ formation. This study concluded that BMP signaling seems to be a critical time-dependent regulator in TMJ development, particularly influencing cell death, specifically highlighted in superficial chondrocytes. This research demonstrates an unanticipated temporal dimension of BMP signaling in TMJ formation, which has differential effects pre- and post- weaning.

A number of weaknesses, however, deserve mention:

The most notable limitation is that, however, they rarely investigated molecular mechanisms. In contrast, though the study clearly shows augmented BMP signaling affects TMJ development, it does not discuss how this occurs at the molecular level. Identifying which of the downstream targets of BMP signaling mediate these effects would greatly strengthen these findings. It remains unknown during post-weaning recovery which compensatory pathways are activated. What triggered this recovery? If the authors include the points in the discussion, the manuscripts will be better. And make their analyses of the clinical implications stronger, please fix some grammar errors.

Answers

We appreciate your valuable comments.

1) For the first comment, we have demonstrated that the BMP/pSMAD pathway upregulates the functions of p53, resulting in augmented cell death in the nasal cartilage and the synchondrosis (Hayano et al., Development, 2015, Ueharu et al., Genesis, 2023). Our another transgenic mice that express constitutive active form of ACVR1 (caAcvr1) in NCCs (P0 Cre; caAcvr1 mice) elevate p53-mediated cell death in the nasal process at the early embryonic stage (Yang et al., J Clin Invest., 2024). Therefore, we consider the BMP/p53 pathway as the molecular mechanism for excess cell death in the TMJ. We have added description below in the Discussion.

Line 196-204: In this study, we have found excess cell death during the TMJ development in a spatial and temporal manner (Figure 4). Our previous study using the same transgenic mouse line demonstrated that enhanced BMP signaling induces p53-mediated cell death in the nasal cartilage and synchondrosis (36, 37). Our other transgenic mouse line to enhance BMP signaling in cranial NCCs (P0 Cre;caAcvr1 mice) also elevates the p53-mediated cell death in the nasal process at the early embryonic stage (38). These data suggest that the excess cell death found in the mutant TMJ is also mediated by p53. However, it is still unclear how the BMP/p53-induced cell death is spatially and temporally regulated during the TMJ development.

2) For the second comment, we suggest an interplay between BMP signaling and mechanosensing. It has been highlighted the importance of the mechanosensing during the TMJ development. We added the description to the discussion part.

Line 220-236: Mechanosensing has recently been highlighted during bone development and craniofacial development. Transcription activators for mechanosensing, YAP and TAZ, govern osteoprogenitor maintenance and subsequent differentiation with BMP signaling (43). Neural crest-specific deletions of Yap and Taz (Wnt1 Cre;Yap+/-;Taz-/- cKO mice) develop ectopic cartilage in the frontal suture leading to premature suture closure (44). Remarkably, our Wnt1 and P0 Cre;caBmpr1a mice also develop ectopic cartilage in the frontal suture that leads to premature suture closure (18). These data suggest that mechanosensing is a critical factor contributing to BMP-induced differentiation toward either osteogenic or chondrogenic lineages depending on the presence or absence of YAP/TAZ. Another aspect is mechanosensing as a conductor for BMP-induced cell death. Neural crest-specific deletion of Bmpr1a reduces cell death in the interzone between the condyle and the glenoid fossa (32), suggesting an essential role of BMP-induced cell death for cavitation in the TMJ. Interestingly, hedgehog signaling, a transducer of mechanosensing, upregulates BMP signaling in TMJ (32, 45). These data suggest that changes in mechanical forces through the transition from the soft diet of milk to hard chow affect either or both cell fate specification and cell death by BMP signaling. 

3) For the third comment, the manuscript was peer-reviewed by scientists whose first language is English to correct any grammatical errors.

Reviewer 2 Report

Comments and Suggestions for Authors

Review comments for ijms-3411019.

In this manuscript, the authors investigated the role of bone morphogenetic protein (BMP) signaling in temporomandibular joint (TMJ) development using conditional knockout mouse models. The authors demonstrated that BMP signaling influences mandibular condyle development through regulation of cell death in superficial chondrocytes, with effects that vary depending on developmental stage. Their histomorphometric analysis revealed reduced cartilage length in pre-weaning stages that recovers post-weaning, suggesting temporal specificity in BMP's influence on TMJ development. While the study presents valuable insights into TMJ development mechanisms, several aspects require attention before publication.

1. Title and Experimental Approach Discrepancy

The manuscript's title refers to "augmented bone morphogenetic protein signaling," yet the experimental approach primarily employs conditional knockout studies that suppress BMP signaling. This misalignment between the title and actual methodology could mislead readers. The authors should revise the title to accurately reflect their experimental approach and findings.

2. Literature Context and Novelty

The relationship between BMP signaling suppression and mandibular growth inhibition has been previously documented in the literature. While this study offers valuable insights through conditional knockout analysis, the authors should more clearly articulate how their findings extend beyond existing knowledge. The discussion section would benefit from a more comprehensive comparison with previous studies to highlight the unique contributions of this work.

3. Methodology Clarification

The methods section would benefit from additional details regarding the selection of developmental timepoints and the rationale behind combining specific zones for analysis. Furthermore, the authors should provide clearer justification for their choice of analytical approaches, particularly in the histomorphometric analysis.

4. Data Interpretation and Physiological Relevance

The recovery of cartilage length post-weaning represents an intriguing finding that warrants deeper discussion. The authors should expand their interpretation of this phenomenon, particularly considering the potential influence of mechanical loading changes during weaning and their physiological significance.

5. Technical Presentation

Minor improvements in figure presentation would enhance the manuscript's clarity. Specifically, the authors should consider adding scale bars consistently across all histological images and providing clearer labeling of the different cellular zones in the relevant figures.

These revisions would strengthen an otherwise well-executed study that provides valuable insights into TMJ development. The suggested modifications are primarily focused on improving clarity and presentation rather than requiring additional experimental work, making this suitable for minor revision.

Author Response

Comments from Reviewer 2

In this manuscript, the authors investigated the role of bone morphogenetic protein (BMP) signaling in temporomandibular joint (TMJ) development using conditional knockout mouse models. The authors demonstrated that BMP signaling influences mandibular condyle development through regulation of cell death in superficial chondrocytes, with effects that vary depending on developmental stage. Their histomorphometric analysis revealed reduced cartilage length in pre-weaning stages that recovers post-weaning, suggesting temporal specificity in BMP's influence on TMJ development. While the study presents valuable insights into TMJ development mechanisms, several aspects require attention before publication.

  1. Title and Experimental Approach Discrepancy

The manuscript's title refers to "augmented bone morphogenetic protein signaling," yet the experimental approach primarily employs conditional knockout studies that suppress BMP signaling. This misalignment between the title and actual methodology could mislead readers. The authors should revise the title to accurately reflect their experimental approach and findings.

Answer

Thank you for your suggestion. However, the title is appropriate because we used transgenic mice that express a constitutive active form of BMP type 1 receptor Bmpr1a (caBmpr1a). By crossing caBmpr1a mice with Wnt1 Cre or P0 Cre mice, we can enhance BMP signaling only in NCCs. We apologize for the confusion. We added the sentences below to the result part to explain the details of our transgenic mice.

Line 67-72: To investigate the impact of augmented BMP signaling in neural crest cells on TMJ development, we crossed transgenic mice that express a constitutively active form of Bmpr1a (caBmpr1a mice) with two widely used neural crest-specific Cre-lines, Wnt1 Cre (22) and P0 Cre (23) to generate Wnt1 Cre;caBmpr1a and P0 Cre;caBmpr1a mutant mice. Both Wnt1 Cre;caBmpr1a mice and P0 Cre;caBmpr1a mice exhibited higher amounts of pSMAD1/5/9 in cranial NCCs compared to controls (18).

  1. Literature Context and Novelty

The relationship between BMP signaling suppression and mandibular growth inhibition has been previously documented in the literature. While this study offers valuable insights through conditional knockout analysis, the authors should more clearly articulate how their findings extend beyond existing knowledge. The discussion section would benefit from a more comprehensive comparison with previous studies to highlight the unique contributions of this work.

Answer

Thank you for your valuable comment. As Reviewer 1 suggested, we have discussed a downstream target of BMP signaling, which leads to excessive cell death in cranial NCCs. Moreover, we also discussed the relation between BMP signaling and mechanosensing during the TMJ development.

Line 196-204: In this study, we have found excess cell death during the TMJ development in a spatial and temporal manner (Figure 4). Our previous study using the same transgenic mouse line demonstrated that enhanced BMP signaling induces p53-mediated cell death in the nasal cartilage and synchondrosis (36, 37). Our other transgenic mouse line to enhance BMP signaling in cranial NCCs (P0 Cre;caAcvr1 mice) also elevates the p53-mediated cell death in the nasal process at the early embryonic stage (38). These data suggest that the excess cell death found in the mutant TMJ is also mediated by p53. However, it is still unclear how the BMP/p53-induced cell death is spatially and temporally regulated during the TMJ development. 

Line 220-236: Mechanosensing has recently been highlighted during bone development and craniofacial development. Transcription activators for mechanosensing, YAP and TAZ, govern osteoprogenitor maintenance and subsequent differentiation with BMP signaling (43). Neural crest-specific deletions of Yap and Taz (Wnt1 Cre;Yap+/-;Taz-/- cKO mice) develop ectopic cartilage in the frontal suture leading to premature suture closure (44). Remarkably, our Wnt1 and P0 Cre;caBmpr1a mice also develop ectopic cartilage in the frontal suture that leads to premature suture closure (18). These data suggest that mechanosensing is a critical factor contributing to BMP-induced differentiation toward either osteogenic or chondrogenic lineages depending on the presence or absence of YAP/TAZ. Another aspect is mechanosensing as a conductor for BMP-induced cell death. Neural crest-specific deletion of Bmpr1a reduces cell death in the interzone between the condyle and the glenoid fossa (32), suggesting an essential role of BMP-induced cell death for cavitation in the TMJ. Interestingly, hedgehog signaling, a transducer of mechanosensing, upregulates BMP signaling in TMJ (32, 45). These data suggest that changes in mechanical forces through the transition from the soft diet of milk to hard chow affect either or both cell fate specification and cell death by BMP signaling.

  1. Methodology Clarification

The methods section would benefit from additional details regarding the selection of developmental timepoints and the rationale behind combining specific zones for analysis. Furthermore, the authors should provide clearer justification for their choice of analytical approaches, particularly in the histomorphometric analysis.

Answers

We appreciate your suggestions. We added the descriptions of the reasons for the developmental timepoints and the justifications for the histomorphometric analysis.

Line 266-267: These timepoints were chosen to assess the progression of TMJ development postnatally with a focus on weaning stages.

Line 272-276: To observe the condyle morphology, Hematoxylin and Eosin (H&E) and Safranin O assays were performed according to standard protocols. Histomorphometric analyses of the condylar cartilage were conducted using ImageJ/Fiji. The mandibular condylar cartilage was divided into superficial (fibrous), proliferative, prehypertrophic, and hypertrophic cell layers based on cell shape, size, and staining intensity (47, 48). 

  1. Data Interpretation and Physiological Relevance

The recovery of cartilage length post-weaning represents an intriguing finding that warrants deeper discussion. The authors should expand their interpretation of this phenomenon, particularly considering the potential influence of mechanical loading changes during weaning and their physiological significance.

Answers: We appreciate your valuable comment. We have added sentences below to the discussion part to expand the potential influence of mechanical loading to the TMJ development.

Line 220-236: Mechanosensing has recently been highlighted during bone development and craniofacial development. Transcription activators for mechanosensing, YAP and TAZ, govern osteoprogenitor maintenance and subsequent differentiation with BMP signaling (43). Neural crest-specific deletions of Yap and Taz (Wnt1 Cre;Yap+/-;Taz-/- cKO mice) develop ectopic cartilage in the frontal suture leading to premature suture closure (44). Remarkably, our Wnt1 and P0 Cre;caBmpr1a mice also develop ectopic cartilage in the frontal suture that leads to premature suture closure (18). These data suggest that mechanosensing is a critical factor contributing to BMP-induced differentiation toward either osteogenic or chondrogenic lineages depending on the presence or absence of YAP/TAZ. Another aspect is mechanosensing as a conductor for BMP-induced cell death. Neural crest-specific deletion of Bmpr1a reduces cell death in the interzone between the condyle and the glenoid fossa (32), suggesting an essential role of BMP-induced cell death for cavitation in the TMJ. Interestingly, hedgehog signaling, a transducer of mechanosensing, upregulates BMP signaling in TMJ (32, 45). These data suggest that changes in mechanical forces through the transition from the soft diet of milk to hard chow affect either or both cell fate specification and cell death by BMP signaling.

  1. Technical Presentation

Minor improvements in figure presentation would enhance the manuscript's clarity. Specifically, the authors should consider adding scale bars consistently across all histological images and providing clearer labeling of the different cellular zones in the relevant figures.

Answer

Thank you for your suggestion. We added scale bars into all histological images (Fig. 2, Fig. 3, and Fig. 4). Figure 2 and 4 have the diagrams of the cellular zones for the histometric analyses.